# Parameter-Free Encoders Remain Viable for RDB Foundation Models

**Linjie Xu** [1]  **David Wipf** [1]

## Abstract

Given a relational database (RDB) storing heterogeneous tabular information, how can we predict missing (or future) values in some target column of interest? As the space of potential targets is vast across enterprise settings, it is preferable to avoid learning a new model from scratch each time there is a new prediction task. Frozen foundation models based on RDB-specific encoders provide a viable solution, but ideal design remains an open question. On the one hand, it has recently been argued that certain *parameter-free* subgraph encoders combined with single-table foundation models can achieve near SOTA performance, with no RDB-specific pre-training required. Meanwhile, other contemporary studies advocate for *parameterized* encoders pre-trained to exploit observable labels for learning task-specific representations. To address this ambiguity, we analyze RDB encoder properties specifically when labels are present as inputs, proving limitations on the potential efficacy of trainable encoder parameters. As empirical validation, we demonstrate that considerably simpler parameter-free encoders are still capable of strong performance across many relevant benchmarking tasks.

## 1. Introduction

Relational databases (RDBs), housing collections of inter-related tables, are indispensable across wide-ranging enterprise applications and e-commerce platforms (Garcia-Molina et al., 2009). From a machine learning standpoint, the information stored within just a single RDB often contains countless possibilities for predictive modeling, covering targets such as customer retention, user churn, or click-through rates (Dave et al., 2014; Motl & Schulte, 2015; Ni et al., 2019; Zykov et al., 2022). Of course retraining a new model from scratch to address each new predictive task of interest may be prohibitively laborious or expensive, which motivates a new class of foundation models (FMs) specif-

ically tailored for RDBs. Among others, these vary from LLM-based approaches that digest serialized RDB content (Wydmuch et al., 2024), to specialized Transformers pre-trained based on in-context learning (ICL) (Fey et al., 2025; Hudovernik et al., 2026; Wang et al., 2026), the common theme in each case being a *frozen* architecture capable of making predictions involving arbitrary unseen RDBs.

As a robust and scalable alternative, it has also been demonstrated that certain *parameter-free* subgraph encoders combined with single-table foundation models can achieve near SOTA performance without requiring any RDB-specific pre-training (Xu et al., 2026). And yet significant ambiguity still exists over RDB embedding and pre-training requirements for FMs. This is largely because of recent architectures with *parameterized* encoders that accept observable neighborhood labels as supplementary discriminative inputs (Chen et al., 2026; Hudovernik et al., 2026; Ranjan et al., 2026), a common technique from graph learning (Shi et al., 2020; Wang et al., 2021). Performance results using such models point towards the continued efficacy of RDB-specific pre-training to optimally exploit these labels on new tasks.

To help resolve this ambiguity, herein we closely examine notable factors influencing the performance of parameterized versus parameter-free RDB encoders, particularly when observable labels are available as encoder inputs. Our contributions are as follows:

- On the theoretical side, we analyze two complementary roles that labels may fill as encoder inputs, either as standard discriminative features or as a mechanism for determining the importance of other discriminative features. This includes proving limitations on the general ability of trainable encoder parameters to exploit these roles w.r.t. downstream performance.

- Consistent with theory, we provide supporting empirical results demonstrating that even when granted labels occupying available inference-time roles, trainable/parameterized encoders do not hold an unequivocal advantage over *much simpler*, parameter-free designs.

## 2. Basics of RDB Foundation Models

The goal of an RDB foundation model is to retain applicability across unseen RDBs, ideally with no retraining required for each new predictive task. Beyond direct LLM-prompting

[1]University of Hong Kong, Shanghai X-Lab. Correspondence to: David Wipf <davidwipf@gmail.com>.

*Proceedings of the $2^{nd}$ ICML Workshop on Foundation Models for Structured Data*, Seoul, South Korea. 2026. Copyright 2026 by the author(s).

approaches (Wydmuch et al., 2024), we now discuss background context and recent pipelines designed for natively handling RDB and/or tabular data.

## 2.1. Predictive Modeling with Relational Context

In traditional supervised learning settings, we are given training data $\mathcal{D} = \{\boldsymbol{X}, \boldsymbol{y}\} \equiv \{\boldsymbol{x}_{i:}, y_i\}_{i=1}^n$ with instance feature rows $\boldsymbol{x}_{i:} \in \mathcal{X}^d$ and corresponding instance labels $y_i \in \mathcal{Y}$ for all $i$. The goal is then to learn a parameterized model $f_\theta$ such that $f_\theta(\boldsymbol{x}_{\text{test}}) \approx y_{\text{test}}$ at any test point $\{\boldsymbol{x}_{\text{test}}, y_{\text{test}}\}$, where in practice $y_{\text{test}}$ is unknown. *RDB predictive modeling* generalizes the above via the inclusion of an additional set of auxiliary data tables $\mathcal{T} = \{\boldsymbol{T}^k\}_{k=1}^K$, where $\boldsymbol{T}^k \in \mathfrak{T}^{n_k \times d_k}$ denotes the $k$-th table associated with a given entity type. Each table row corresponds with a single instance of that entity (e.g., an individual user), and the columns encode heterogeneous instance attributes (e.g., elements of a user profile). Note that without loss of generality, we also assert that $\boldsymbol{T}^K \equiv \boldsymbol{X}$. To complete its specification and make use of these auxiliary tables for making predictions, an RDB also includes a set of relations $\mathcal{R}$ expressed as primary key (PK) foreign key (FK) pairs (Garcia-Molina et al., 2009).

## 2.2. Computing RDB Embeddings

To make use of relational context for predictive tasks, the basic strategy is to first convert a given RDB $\{\mathcal{T}, \mathcal{R}\}$ to a heterogeneous graph $\mathcal{G}$. This typically involves treating each table $\boldsymbol{T}^k$ as a node type and each row $i$ within a given $\boldsymbol{T}^k$ as a node with features $\boldsymbol{t}_{i:}^k$ (Cvitkovic, 2020; Dwivedi et al., 2025; Fey et al., 2023; Zhang et al., 2023a;b). Directed edges can then be formed using each PK-FK pair within $\mathcal{R}$. We next sample $H$-hop subgraphs or ego-networks $\mathcal{G}_H(\boldsymbol{x}_{i:})$ from $\mathcal{G}$ that are centered at each target row $\boldsymbol{x}_{i:}$ within $\boldsymbol{X}$. Importantly, sampling for *temporal* RDBs should exclude nodes with time-stamps later than $\boldsymbol{x}_{i:}$. These subgraphs form the input to an RDB encoder $g_\phi$, instantiated as some form of graph neural network (GNN) or graph Transformer architecture with (optional) parameters $\phi$. This facilitates computation of *fixed-length* embeddings $\boldsymbol{z}_{i:} = g_\phi[\mathcal{G}_H(\boldsymbol{x}_{i:})] \in \mathbb{R}^{d_z}$.

## 2.3. Combining with RDB Prediction Heads

Embeddings obtained via $g_\phi$ are combined with a parameterized prediction head $q_\theta$ in two primary ways.

**Schema-agnostic models.** Provided the encoder $g_\phi$ is designed to digest a broad spectrum of input RDB schema using a shared representational form, then it is possible to train a single composite model $q_\theta(y \mid \boldsymbol{z}_{\text{test}}) \equiv q_\theta(y \mid g_\phi[\mathcal{G}_H(\boldsymbol{x}_{\text{test}})])$ to assign high probability to labels $y = y_{\text{test}}$ (or $y \approx y_{\text{test}}$ for regression tasks) across a range of real-world RDBs (Ranjan et al., 2026; Wang et al., 2025; Wu et al., 2025). Here $\boldsymbol{x}_{\text{test}}$ can be viewed as additional unlabeled rows appended to $\boldsymbol{X}$. Such models can in principle be applied to new unseen RDBs without further training.

**ICL-based models.** A limitation of the schema-agnostic models from above is that the effective context available for making predictions is merely the subgraph $\mathcal{G}_H(\boldsymbol{x}_{\text{test}})$, which in isolation may not contain sufficient information pertaining to broader labeling patterns across a vast RDB. In-context learning provides a natural workaround by granting the prediction head diverse samples of multiple analogous subgraphs, including past labels where available, extracted from each new RDB of interest. Specifically, ICL-based architectures derive predictions from the revised

$$q_\theta\big(y \mid \boldsymbol{z}_{\text{test}}, \mathcal{D}\big), \text{ with } \mathcal{D} = \{\boldsymbol{z}_{i:}, y_i\}_{i=1}^n, \ \boldsymbol{z} = g_\phi[\mathcal{G}_H(\boldsymbol{x})],$$

where $\boldsymbol{z}_{\text{test}}$ and each $\boldsymbol{z}_{i:}$ are obtained by passing unlabeled and labeled rows of $\boldsymbol{X}$ through the shared encoder $g_\theta$. Mirroring a growing number of successful single-table foundation model designs (Hollmann et al., 2025; Jingang et al., 2025; Qu et al., 2026; Zhang et al., 2025a;b), $q_\theta$ is implemented via various Transformer architectures that grant flexibility over $n$ and $d_z$. Finally, model pre-training is conducted using large volumes of synthetically-generated data following single-table pipelines, possibly combined with real-world RDBs (Chen et al., 2026; Fey et al., 2025; Hudovernik et al., 2026; Kothapalli et al., 2026).

## 2.4. Two Foundational Questions

**Is RDB-Specific Pre-Training Actually Necessary?** It is initially reasonable to assume that pre-training RDB foundation models specifically using RDB data is absolutely essential for achieving strong inference-time performance. However, recent work (Xu et al., 2026) has suggested otherwise for ICL-based architectures. The high-level idea is that prior to seeing ICL samples, a necessarily *fixed* $g_\theta$ is incapable of adjudicating column roles, which may vary from dataset to dataset (or even task to task within a given RDB). This phenomena advantages RDB encoders that only compress vertically *within* high-dimensional columns (where shared units facilitate interpretable aggregation), not horizontally *across* columns, where relevance is largely indeterminate adequate label information. Conditioned on this restriction to vertical compression, Xu et al. (2026) formalize how a *parameter-free* encoder results in a minimal loss of expressiveness. And via the associated *RDBLearn* toolbox, we can directly pair this class of encoder with the most powerful existing single-table foundation models such that no actual RDB-specific pre-training is required. Empirically, this strategy is shown to match or exceed alternative RDB foundation models on previously-unseen predictive tasks.

**Does providing the encoder with labels impact pre-training requirements?** Two counter-points challenge the efficacy of parameter-free encoders upon which the training-free approach mentioned above is based. Both involve the use of observable labels as inputs to a trainable $g_\phi$. We analyze these scenarios next, providing new insights that inform the foundational questions of this section.

# 3. Labels as Extra Discriminative Features

Per previously-stated definitions, an RDB neighborhood subgraph $\mathcal{G}_H(\boldsymbol{x})$ may include nodes associated with other rows of $\boldsymbol{T}^K \equiv \boldsymbol{X}$, provided these rows are reachable within $H$ PK-FK hops and have an earlier time-stamp to avoid temporal leakage. As these rows have associated labels, it is natural to consider including them as extra node features to improve downstream predictability. We adopt $\mathcal{G}_H^*(\boldsymbol{x})$ to reference the resulting subgraph with these labels concatenated as additional node features; we are assuming here that any labels included within $\mathcal{G}_H^*(\boldsymbol{x})$ have time-stamps earlier than $\boldsymbol{x}$. In more traditional supervised learning regimes involving graphs, this is a common practice for improving node classification accuracy (Wang et al., 2022) (akin to a trainable form of label propagation). More recently, analogous ideas have been applied to RDB foundation models, whereby trainable encoders are explicitly designed to process past labels of similar entities within an RDB neighborhood (Chen et al., 2026; Hudovernik et al., 2026; Ranjan et al., 2026).

But to what degree can we actually rely on such neighborhood labels as predictive features, specifically in the context of trainable RDB foundation model encoders $g_\phi$ applied to $\mathcal{G}_H^*(\boldsymbol{x}_{\text{test}})$ evaluated at an unlabeled test row of interest $\boldsymbol{x}_{\text{test}}$. To begin, for an arbitrary RDB we define $\mathcal{G}_{\text{test}}^*$ as an augmented version of $\mathcal{G}$, with $\boldsymbol{x}_{\text{test}}$ appended to $\boldsymbol{X}$, known labels associated with $\boldsymbol{X}$ included as candidate node features, but restricted to entities with timestamps earlier than $\boldsymbol{x}_{\text{test}}$. We also let $\widetilde{\mathcal{G}}_H^*(\boldsymbol{x}_{\text{test}})$ denote $\mathcal{G}_{\text{test}}^* \backslash \mathcal{G}_H^*(\boldsymbol{x}_{\text{test}})$, meaning all RDB context outside of the label-infused $H$-hop ego-network centered at test point $\boldsymbol{x}_{\text{test}}$. And lastly, we adopt $y_H(\boldsymbol{x}_{\text{test}})$ to indicate the set of all past labels present within $\mathcal{G}_H^*(\boldsymbol{x}_{\text{test}})$; this of course does *not* include the unknown label $y_{\text{test}}$ associated with $\boldsymbol{x}_{\text{test}}$ itself. We then have the following:

**Proposition 3.1.** *Given any ego-network $\mathcal{G}_H^*(\boldsymbol{x}_{\text{test}})$ with $|y_H(\boldsymbol{x}_{\text{test}})| > 0$, distributions over remaining RDB content $p\big(\widetilde{\mathcal{G}}_H^*(\boldsymbol{x}_{\text{test}}), y_{\text{test}}|\mathcal{G}_H^*(\boldsymbol{x}_{\text{test}})\big)$ will always exist such that:*

1. *(Foundation model setting) $y_{\text{test}}$ is a deterministic function of $\mathcal{G}_{\text{test}}^*$, and yet any possible fixed encoder $g_\phi$ and prediction head $q_\theta$ perform no better than chance estimating $y_{\text{test}}$ in the sense that*

$$\mathbb{E}\Big[\mathbb{KL}\Big[p(y_{\text{test}}|\mathcal{G}_{\text{test}}^*) \,||\, q_\theta\big(y_{\text{test}} \mid g_\phi\big[\mathcal{G}_H^*(\boldsymbol{x}_{\text{test}})\big]\big)\Big]\Big] \quad (1)$$
$$\geq \mathbb{E}\Big[\mathbb{KL}\Big[p(y_{\text{test}}|\mathcal{G}_{\text{test}}^*) \,||\, \text{Unif}\big(y_{\text{test}}\big)\Big]\Big],$$

*with outer expectations over $p\big(\widetilde{\mathcal{G}}_H^*(\boldsymbol{x}_{\text{test}})|\mathcal{G}_H^*(\boldsymbol{x}_{\text{test}})\big)$.*

2. *(Supervised setting) Given $\{\mathcal{G}_H^*(\boldsymbol{x}_{i:}), y_i\}_{i=1}^n$ and $y_{\text{test}}$ extracted from any single draw from $p\big(\widetilde{\mathcal{G}}_H^*(\boldsymbol{x}_{\text{test}}), y_{\text{test}}|\mathcal{G}_H^*(\boldsymbol{x}_{\text{test}})\big)$, there exists a draw-specific $f_\theta = q_\theta \circ g$ satisfying $y_i = f_\theta\big[\mathcal{G}_H^*(\boldsymbol{x}_{i:})\big]$ for all $i$ and $y_{\text{test}} = f_\theta\big[\mathcal{G}_H^*(\boldsymbol{x}_{\text{test}})\big]$, where $g$ is a parameter-free mapping from any $\mathcal{G}_H^*(\boldsymbol{x}_{\text{test}})$ to $\mathbb{R}^m$ and $q_\theta$ is a ReLU network with finite width and depth.*

**Intuition behind of Proposition 3.1.** Neighborhood labels within $\mathcal{G}_H^*(\boldsymbol{x})$ can be highly discriminative in a *supervised* setting, where training specifically on samples $\{\mathcal{G}_H^*(\boldsymbol{x}_{i:}), y_i\}_{i=1}^n$ extracted from a *single* RDB predictive task facilitate learning a task-specific discriminative function $f_\theta$ applicable to test points $\mathcal{G}_H^*(\boldsymbol{x}_{\text{test}})$ (part 2). And yet it is not generally possible to determine a stable predictive role for these labels within an encoder that must remain *fixed* across varying datasets/tasks facing a foundation model architecture of the form $q_\theta\big(y \mid g_\phi\big[\mathcal{G}_H^*(\boldsymbol{x}_{\text{test}})\big]\big)$ (part 1). Given this implication of Proposition 3.1, why then are existing schema-agnostic foundation models like the relational Transformer (RT) model from Ranjan et al. (2026) capable of making successful predictions? A simple plausible hypothesis is that, within the handful of real-world RDBs available for pre-training RT models, homophily relationships[1] dominate, so simply adopting the majority label of local neighbors can be a highly discriminative feature. Empirical results from Ranjan et al. (2026)[Table 1] support this conclusion, namely, the similar performance of a basic entity mean predictor explicitly predicated on homophily (66.7 avg AUROC) relative to RT (69.7 avg AUROC).

**Implications for ICL-based models.** ICL provides one pathway for diversifying label usage effectively beyond the homophily-dominant regimes mentioned above; however, this remains true whether or not a parameter-free encoder is adopted. This is because even without trainable parameters, encoder representations can include diverse label aggregations that are sensitive to both homophily and heterophily relationships; the subsequent ICL-prediction head then sorts which are relevant for any given task. Note that canonical aggregators can trivially handle the parity check example constructed within the proof of Proposition 3.1 (see Appendix D). Hence given evidence available thus far, the use of labels as discriminative features does not by itself provide a compelling reason to favor trainable parameterized RDB encoders within ICL-based models.

# 4. Labels for Adjudicating Feature Importance

Analysis from Xu et al. (2026) indicates that uninformative feature columns within RDBs cannot generally be filtered out using a dataset-agnostic encoder, which reduces the motivation for including trainable encoder parameters. However, follow-up study (Hudovernik et al., 2026) suggests that this calculus may change if the encoder is provided with local neighborhood labels (as in $\mathcal{G}_H^*(\boldsymbol{x})$). The latter provide task-specific information, which may plausibly facilitate feature-column differentiation on a task-by-task basis as required by RDB foundation models. But is such differen-

---

[1]In the context of heterogeneous graphs or subgraphs defining RDB relations, homophily refers to the tendency of neighboring nodes of the same node type (not necessarily 1-hop neighbors) sharing the same label, while heterophily represents the converse.

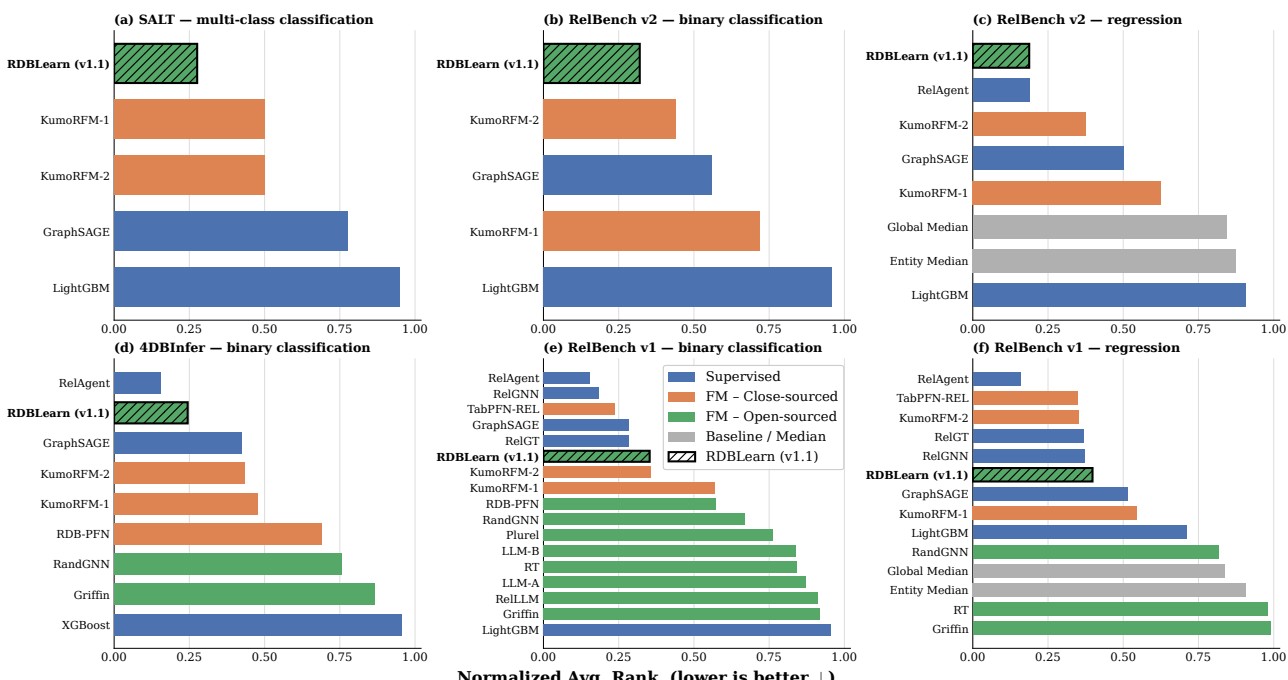

*Figure 1.* RDBLearn outperforms all open-source FMs on all benchmarks, and even exceeds closed-source FMs on 4 of 6 benchmarks.

tiation generally possible even when $\mathcal{G}_H(\boldsymbol{x})$ is switched to $\mathcal{G}_H^*(\boldsymbol{x})$? The answer is negative in the following sense:

**Proposition 4.1.** *Given any representative RDB ego-network $\mathcal{G}_H(\boldsymbol{x}_{test})$ constructed from non-empty tabular feature columns and $|y_H(\boldsymbol{x}_{test})| > 0$, there will always exist distributions over $\widetilde{\mathcal{G}}_H^*(\boldsymbol{x}_{test})$ and $y_H(\boldsymbol{x}_{test}) \cup y_{test}$ such that:*

1. *Each label in $y_H(\boldsymbol{x}_{test}) \cup y_{test}$ is a shared deterministic function of some support set $S$ of tabular column features and independent of all remaining columns.*

2. *Encoder input labels $y_H(\boldsymbol{x}_{test})$ provide no information about the unknown generating support set $S$ in the sense that $p\big(S|\mathcal{G}_H^*(\boldsymbol{x}_{test})\big) = p\big(S|\mathcal{G}_H(\boldsymbol{x}_{test})\big).$*

**Implications for ICL-based models.** While not strictly ruling out any benefit of trainable encoders, Proposition 4.1 does mute confidence in their ability to robustly screen uninformative columns through the use of local neighborhood labels as additional inputs. This plausibly suggests that, at least for RDBs with large numbers of feature columns (many of which are likely uninformative), trainable encoders may be unnecessary, especially given that parameter-free encoders are already structured to naturally handle column-wise feature screening (Xu et al., 2026). Empirical results presented next further support these observations.

## 5. Empirical Support

**Models.** We adopt an updated version of *RDBLearn* (Zhang et al., 2026), the RDB FM pipeline with parameter-free encoder as analyzed by Xu et al. (2026); see Appendix

A for our RDBLearn update details. For parameterized encoder FMs, we include a variety of both open-source and closed-source options with published results based on author specific tuning. Note that most models only provide results on a subset of available benchmarks. Lastly, we cover a handful of supervised baselines that represent either SOTA performance or classical methodology as a reference point. Appendix B includes all baseline model descriptions.

**Results.** We compare across a suite of six diverse benchmarks from Appendix B, each composed of multiple RDB predictive tasks. Results are shown in Figure 1 and Appendix C, where RDBLearn (with just a simple parameter-free encoder) achieves the best overall performance among open-source FMs. And with the exception of two RelBench-v1 benchmarks, RDBLearn even outperforms all closed-source FMs that may involve pre-training with similarly-structured RDB content. We stress that RDBLearn was run directly from the open-source repository for all benchmarks with no task-specific adjustments and zero exposure to RDB data of any kind at any stage, i.e., there is no chance for any implicit test-set alignment on limited public benchmarks.

## 6. Take-Home Message

Our empirical results indicate that, at least until further advancements in the field, parameter-free RDB encoding combined with frontier single-table FMs is still competitive with the best closed-source industry architectures equipped with pre-trained, parameterized encoders. These findings are further supported by theoretical study of the encoder representations possible with neighborhood labels.

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

# A. RDBLearn Updates

We describe a few modest updates to the RDBLearn package, which collectively constitute what we henceforth refer to as **RDBLearn version 1.1**.[2] These include the following:

- **Aggregations:** Because the original RDBLearn package (Zhang et al., 2026) only supports a minimal set of rudimentary aggregation functions for encoding purposes (mean, min/max, standard deviation, counts, median), we explore a broader set that includes 25-th quantiles and 75-th quantiles for continuous feature columns, as well as discrete entropy for categorical feature columns. These increase model expressivity and are shared across all tasks/datasets. We also add the option to treat categorical variables as numerical values, which alters the space of candidate aggregators. When available, the validation set performance is used to select between the original aggregations and the expanded group listed here. See below for handling instances with no validation set.

- **Base Model Predictors:** One of the advantages of the RDBLearn architecture is that we may seamlessly incorporate any new frontier single-table foundation model. For RDBLearn-v1.1 we add TabICL-v2 (Qu et al., 2026) and TabPFN-v3 (Grinsztajn et al., 2026).

- **Default Setting:** For any benchmark with no official validation set, we adopt a simplified RDBLearn default setting. Specifically, the base tabular predictor is fixed as TabPFNv2.5 (Grinsztajn et al., 2025), encoder depth is $H = 4$, and aggregation is the new expanded set mentioned above.

We remark that there are multiple promising directions whereby RDBLearn could plausibly be improved further. For example, presently RDBLearn simply ignores all text-based columns, unlike most other foundation models that adopt some form of text embedding. And secondly, RDBLearn does not as of yet provide its encoder with any local neighborhood labels, the topic of Sections 3 and 4. Note that even though we have argued in the main text that the presence of such labels need not necessarily favor a parameterized encoder over a parameter-free one, our analysis does *not* suggest that such labels (when available) possess no advantage as extra encoder inputs. In fact, there may well be many instances where their inclusion directly benefits parameter-free encoder models like RDBLearn. Certainly prior analysis from elsewhere indicates that they have discriminative value in broader contexts (Wang et al., 2022).

# B. Experiment Details

### B.1. Description of Baseline Approaches

We present evaluations across a wide range of RDB predictive architectures, including both foundation models and more traditional supervised learning approaches, although results are not presently available for all models on all datasets. For a detailed taxonomy of the FM architectures, please see Figure 2.

**Closed-Source Foundation Models.** We include the original industry model **KumoRFM-1** (Fey et al., 2025) as well as the more recent version **KumoRFM-2** (Hudovernik et al., 2026) that incorporates task-specific labels to the encoder as discussed in Section 4 of our main text. We also consider **TabPFN-REL**, which is likely based at least in part on the recently-released TabPFN-v3 (Grinsztajn et al., 2026), although no RDB-specific details of the architecture have been disclosed as of yet. Lastly, we compare with **OpenRFM** (Chen et al., 2026); while this model is presumably intended to be open-sourced at some point, presently the full code is not available. For all of these closed-source approaches we rely on posted performance results from original authors for evaluation purposes, with one exception. We follow Grinsztajn et al. (2026), who have independently reevaluated KumoRFM-2 performance using API calls for certain datasets using author-provided benchmarking scripts; see further details in Appendix C.

**Open-Source Foundation Models.** We compare RDBLearn against a variety of open-source foundation models including the schema-agnostic **Griffin** (Wang et al., 2025), **RelLLM** (Wu et al., 2025), and **RT** (relational transformer) (Ranjan et al., 2026) models, as well as an updated RT version enhanced with **PluRel** synthetic pretraining data (Kothapalli et al., 2026). On the open-source ICL-based side we include results from **RDB-PFN** (Wang et al., 2026) and our own implemented baseline **RandGNN**, which combines a random-weight GNN encoder with a single-table FM (see Xu et al. (2026) for details). RandGNN represents a form of ablation w.r.t. RDBLearn, in that it explicitly mixes cross-column information in a manner at odds with the theoretical considerations upon which RDBLearn is based. And finally, for pure language model baselines, we evaluate **LLM-A** (Team et al., 2025) (as tested in Ranjan et al. (2026)) and **LLM-B** (Wydmuch et al.,

---

[2]https://github.com/HKUSHXLab/rdblearn/releases/tag/v1.1

2024); both approaches are predicated on serialized RDB neighborhood representations and/or ICL samples. Although the underlying LLMs are not open-source, we nonetheless categorize these two approaches as open-source in that the prompting needed to reproduce results are publicly available and no hidden fine-tuning steps are involved.

| | open-source | parameter-free encoder | RDB pretraining required | real RDB exposure | LLM usage | ICL-based |
|---|---|---|---|---|---|---|
| **KumoRFM-1** | N | N | Y | Y | ? | Y |
| **KumoRFM-2** | N | N | Y | Y | ? | Y |
| **TabPFN-REL** | N | ? | ? | ? | ? | ? |
| **OpenRFM** | N* | N | Y | Y | N | Y |
| **LLM-A** | Y | N/A | N | N | Y | Y |
| **LLM-B** | Y | N/A | N | N | Y | Y |
| **RelLLM** | Y | N | Y | Y | Y | N |
| **Griffin** | Y | N | Y | Y | N | N |
| **RT** | Y | N | Y | Y | N | N |
| **PluRel** | Y | N | Y | Y** | N | N |
| **RDB-PFN** | Y | Y | Y | N | N | Y |
| **RandGNN** | Y | N | N | N | N | Y |
| **RDBLearn** | Y | Y | N | N | N | Y |

*Figure 2. Taxonomy of existing RDB foundation models.* For closed-source models, not all attributes can be determined from limited publicly-available information. Additionally, pure LLM-based approaches (namely LLM-A and LLM-B) do not have an RDB-specific encoder per se, hence we select N/A = not applicable for this category.
* We categorize OpenRFM as a closed-source model at the time of this writing because code is not presently available for running the model, nor does there exist a sufficient published description for full implementation.
** Although the PluRel paper (Kothapalli et al., 2026) considers models both with and without real-world RDB data for pre-training, performance of the synthetic-only version is weaker and not included in our comparisons.

**Task-Specific Supervised Models.** While our emphasis is on RDB foundation models, it is still useful to contextualize FM performance results w.r.t. methods benefitting from per-task supervised training. On this front, largely following prior work, we include the more classical standards **LightGBM**, **XGBoost**, and **GraphSage** as evaluated in Hudovernik et al. (2026). Meanwhile, for recent SOTA supervised architectures we select **RelGNN** (Chen et al., 2025), **RelGT** (Dwivedi et al., 2025), and **RelAgent** (Huang et al., 2026).

### B.2. RDB Benchmarks

We evaluate RDB predictive modeling approaches over a diverse suite of benchmarks, including regression and binary classification tasks from both RelBench-v1 (Robinson et al., 2024) and RelBench-v2 (Gu et al., 2026). Beyond these, we test over the 4DBInfer (Wang et al., 2024) binary classification datasets used in several recent works, as well as the multi-class predictive tasks from the SALT repository (Klein et al., 2024) as adopted by Hudovernik et al. (2026). Note that for RelBench-v1, we omit only rel-f1 and rel-event datasets because of potential leakage issues mentioned elsewhere.

### B.3. Subtleties of RDB Benchmarking

**Language Model Agents.** A growing body of evidence suggests that LLM agents can play a useful role in tabular model development (Han et al., 2024; Huang et al., 2026; Kim et al., 2026; Zhang et al., 2024). That being said, their usage within the context of RDB benchmarking involves notable caveats. One in particular relates to potential leakage issues that may obscure the degree to which we can plausibly extrapolate benchmarking results to expected performance within siloed industry settings. This concern stems from the fact that existing public benchmarks are scarce, largely because of privacy concerns with real-world data. And yet predictive solutions addressing the few RDB datasets that do exist, often involving feature engineering and human expertise, are readily visible to frontier LLM agents during their pre-training process. Hence agent-based solutions may simply lean on problem-specific prior human knowledge to a degree that may not be possible on less-familiar or esoteric tasks within private RDB repositories. Of course this is not meant to discourage LLM-based RDB frameworks, and certainly RDBLearn is a natural fit for agentic enhancements (Xu et al., 2026). We only advocate that some caution is warranted when interpreting results, especially when extrapolating to new domains. While similar concerns have certainly been raised with other data modalities, we would argue that the situation is more acute in RDB regimes with only limited, relatively well-known benchmarking sources that are susceptible to memorized solutions.

**Validation Set Usage within Foundation Models.** Hudovernik et al. (2026) mention that as an RDB foundation model, KumoRFM-2 does not require a validation set for task-specific training. Consequently, for reported results they merge official training-set and validaion-set splits for constructing ICL samples. This can provide an advantage, since for temporal datasets

(the norm with RDBs), validation-set splits are often closer in time to the test set. But this same strategy can be applied to other RDB foundation models, including parameter-free encoder architectures like RDBLearn. In preliminary testing with this approach, we observe that RDBLearn can indeed incur benefits, although further study is warranted to explore how to balance training-set and validation-set contributions within ICL sample batches. We also note that KumoRFM-2 usage depends on a broad set of explicit hyperparameters, the default settings of which vary across tasks and datasets according to author-provided scripts.[3] However, it remains unclear how these were chosen, e.g., using validation set performance (which might entangle with the ICL-sample selection mentioned above) or else via some other undisclosed criteria. Without such a prescription, it is also uncertain how these hyperparameters should be adapted for new user-provided RDB predictive tasks.

## C. Extended Empirical Results

In this section we provide the full set of experimental results upon which the final rankings shown in Figure 1 from the main text are based. Overall, RDBLearn is consistently competitive against other approaches that possess more sophisticated pre-trained parameterized encoders. Per available results thus far, RDBLearn is superior to all foundation models on all tasks, with the exception of only the closed-source TabPFN-REL, and conditionally KumoRFM-2 (see Tables 5 and 6 below), restricted to RelBench-v1 datasets. We remark that many enhancements potentially adopted by these models (e.g., intelligent ensembling, post-processing, ICL sampling strategies, and/or text-field handling as mentioned by Hudovernik et al. (2026)) could equally-well benefit RDBLearn, and need not undermine the utility of a parameter-free encoder design.

*Table 1.* Results on 4DBInfer binary classification tasks (AUC↑). Bold = best per column; underline = second best.

| Method | AB churn | OB ctr | RR cvr | SE upvote | SE churn | Avg Rank ↓ |
|---|---|---|---|---|---|---|
| **Supervised** | | | | | | |
| XGBoost | 50.00 | 50.00 | 50.00 | 49.68 | 50.84 | 8.6 |
| GraphSage | 75.71 | 62.39 | 84.70 | 88.61 | 85.58 | 3.8 |
| RelAgent | **79.44** | 62.63 | 86.35 | **89.45** | **89.03** | **1.4** |
| **Foundation model** | | | | | | |
| *Closed-Source* | | | | | | |
| KumoRFM-1 | 75.59 | 61.11 | 84.36 | 88.92 | 87.87 | 4.3 |
| KumoRFM-2 | 77.06 | 61.97 | 84.36 | 88.50 | 87.92 | 3.9 |
| RDB-PFN | 73.38 | 52.59 | 78.12 | 85.60 | 85.50 | 6.2 |
| *Open-Source* | | | | | | |
| Griffin | 53.32 | 52.22 | 39.92 | 71.66 | 72.18 | 7.8 |
| RandGNN | 77.42 | 49.29 | 66.21 | 61.06 | 77.41 | 6.8 |
| RDBLearn (v1.1) | 78.50 | **63.76** | **89.06** | 87.86 | 88.69 | 2.2 |

*Table 2.* Results on the SALT benchmark (MRR↑). Bold = best per column; underline = second best.

| Method | salt-item | | | salt-sales | | | | | Avg MRR | Avg Rank |
| | Plant | Shipping Point | Incoterm | Incoterm | Office | Group | Payment Terms | Shipping Condition | ↑ | ↓ |
|---|---|---|---|---|---|---|---|---|---|---|
| **Supervised** | | | | | | | | | | |
| GraphSAGE | 0.99 | 0.98 | 0.69 | 0.62 | **1** | 0.16 | 0.37 | 0.57 | 0.67 | 3.88 |
| LightGBM | 0.61 | 0.28 | 0.73 | 0.73 | 0.99 | 0.02 | 0.1 | 0.51 | 0.50 | 4.75 |
| **Foundation model** | | | | | | | | | | |
| *Closed-Source* | | | | | | | | | | |
| KumoRFM-1 | 0.99 | **0.99** | 0.79 | 0.81 | **1** | 0.38 | 0.38 | 0.78 | 0.77 | 2.50 |
| KumoRFM-2 | 0.99 | 0.98 | 0.78 | 0.81 | **1** | 0.61 | 0.61 | 0.79 | 0.82 | 2.50 |
| *Open-Source* | | | | | | | | | | |
| RDBLearn (v1.1) | 1 | 0.99 | 0.79 | 0.9 | 1 | 0.61 | 0.75 | 0.83 | **0.86** | **1.38** |

---

[3] See https://github.com/kumo-ai/kumo-rfm/tree/master/benchmarks/v2.

*Table 3.* Results on Relbench v2 binary classification tasks (AUROC↑). Bold = best per column, underline = second best.

| Method | mimic stay | ratebeer beer-churn | ratebeer user-churn | ratebeer dormant | arxiv citation | Avg AUROC ↑ | Avg Rank ↓ |
|---|---|---|---|---|---|---|---|
| **Supervised** | | | | | | | |
| LightGBM | 51.81 | 76.21 | 83.92 | 75.79 | 71.21 | 71.79 | 4.80 |
| GraphSAGE | 55.01 | 78.67 | 94.27 | 80.51 | 82.50 | 78.19 | 2.80 |
| Rel-Agent[†] | – | **84.70** | 97.43 | **83.33** | **82.61** | – | – |
| **Foundation model** | | | | | | | |
| *Closed-Source* | | | | | | | |
| KumoRFM-1 | **56.33** | 75.06 | 91.38 | 77.10 | 80.62 | 76.10 | 3.60 |
| KumoRFM-2 | 55.28 | 83.84 | 97.43 | 80.65 | 81.71 | 79.78 | 2.20 |
| OpenRFM (tabicl)[†] | – | 79.60 | – | 77.50 | 81.00 | – | – |
| OpenRFM (tabpfn)[†] | – | 80.90 | – | 79.10 | 81.00 | – | – |
| *Open-Source* | | | | | | | |
| RDBLearn | 55.40 | 83.56 | **97.84** | 81.18 | 82.37 | **80.07** | **1.60** |

[†] Rel-Agent and OpenRFM are excluded when calculating average AUROC and Rank because results are missing in some cases (Rel-Agent: no mimic-stay; OpenRFM: no mimic-stay or ratebeer user-churn). Note that public code is not available to produce results for the missing tasks.

*Table 4.* Results on Relbench-v2 regression tasks (MAE↓). Bold = best per column; underline = second best.

| Method | ratebeer user-count | arxiv publication | Avg MAE ↓ | Avg Rank ↓ |
|---|---|---|---|---|
| **Baseline** | | | | |
| Global Median | 15.124 | 0.577 | 7.851 | 6.75 |
| Entity Median | 13.079 | 0.874 | 6.977 | 7.00 |
| **Supervised** | | | | |
| LightGBM | 20.350 | 0.577 | 10.464 | 7.25 |
| GraphSAGE | 7.374 | 0.513 | 3.944 | 4.00 |
| RelAgent | **6.021** | 0.462 | **3.242** | **1.50** |
| **Foundation model** | | | | |
| *Closed-Source* | | | | |
| KumoRFM-1 | 11.063 | 0.518 | 5.791 | 5.00 |
| KumoRFM-2 | 7.298 | 0.487 | 3.893 | 3.00 |
| *Open-Source* | | | | |
| RDBLearn (v1.1) | 6.870 | **0.459** | 3.665 | **1.50** |

*Table 5.* Results on Relbench v1 binary classification tasks (AUROC↑). Bold = best per column, underline = second best.

| | avito | | trial | amazon | | stack | | hm | Avg AUROC | Avg Rank |
|---|---|---|---|---|---|---|---|---|---|---|
| **Method** | click | visit | out | user | item | eng | badge | churn | ↑ | ↓ |
| **Supervised** | | | | | | | | | | |
| LightGBM | 53.60 | 53.05 | 70.09 | 52.22 | 62.54 | 63.39 | 63.43 | 55.21 | 59.19 | 16.25 |
| GraphSAGE | 65.90 | 66.20 | 68.60 | 70.42 | 82.81 | 90.59 | 88.86 | 69.88 | 75.41 | 4.81 |
| RelGT | 68.30 | 66.78 | 68.61 | 70.39 | 82.55 | 90.53 | 86.32 | 69.27 | 75.34 | 4.81 |
| RelGNN | 68.23 | 66.18 | 71.24 | **70.99** | 82.64 | **90.75** | **88.98** | 70.93 | 76.24 | 3.12 |
| RelAgent | **68.36** | 67.79 | 71.86 | 70.78 | **82.84** | 90.41 | 88.42 | **71.07** | **76.44** | **2.62** |
| **Foundation model** | | | | | | | | | | |
| *Closed-Source* | | | | | | | | | | |
| KumoRFM-1 | 64.85 | 64.11 | 70.79 | 67.29 | 79.93 | 87.09 | 80.00 | 67.71 | 72.72 | 9.69 |
| KumoRFM-2 (orig.)[*] | 67.42 | **69.41** | 72.03 | 69.10 | 82.17 | 89.40 | 87.15 | 69.27 | 75.74 | 5.00 |
| KumoRFM-2 (repro.)[*] | 67.42 | **69.41** | 72.03 | 67.71 | 80.18 | 88.69 | 85.40 | 67.81 | 74.83 | 6.06 |
| TabPFN-REL | 67.09 | 66.68 | **76.43** | 70.27 | 82.81 | 90.66 | 85.17 | 70.55 | 76.21 | 4.06 |
| OpenRFM (TabICL)[†] | 62.30 | 64.60 | 60.80 | – | – | 88.50 | 86.30 | 68.60 | – | – |
| OpenRFM (TabPFN)[†] | 63.90 | 64.60 | 63.40 | – | – | 88.40 | 86.40 | 68.90 | – | – |
| *Open-Source* | | | | | | | | | | |
| LLM-A | 59.80 | 62.70 | 57.40 | 58.10 | 62.10 | 78.00 | 80.00 | 59.80 | 64.74 | 14.81 |
| LLM-B | 61.32 | 60.28 | 55.72 | 60.56 | 71.96 | 81.01 | 71.13 | 64.34 | 65.79 | 14.25 |
| RelLLM | 62.28 | 56.17 | 59.02 | 60.07 | 64.10 | 69.46 | 62.12 | 55.95 | 61.15 | 15.50 |
| Griffin | 45.90 | 60.70 | 51.00 | 62.30 | 69.00 | 77.50 | 73.50 | 60.20 | 62.51 | 15.62 |
| RT | 59.50 | 61.80 | 51.80 | 64.00 | 70.90 | 75.70 | 80.10 | 62.80 | 65.83 | 14.31 |
| Plurel | 47.90 | 63.40 | 51.80 | 65.00 | 72.50 | 86.20 | 82.00 | 66.00 | 66.85 | 12.94 |
| RDB-PFN | 64.46 | 65.82 | 64.50 | 65.92 | 79.59 | 88.04 | 84.52 | 67.21 | 72.51 | 9.75 |
| RandGNN | 61.58 | 64.15 | 55.98 | 65.87 | 77.73 | 85.14 | 83.14 | 66.54 | 70.02 | 11.38 |
| RDBLearn (v1.1) | 67.55 | 66.00 | 72.71 | 68.81 | 82.25 | 89.98 | 84.62 | 68.22 | 75.02 | 6.00 |

[*] Results are drawn from both the original KumoRFM-2 paper (Hudovernik et al., 2026), as well as an independent reproduction from Grinsztajn et al. (2026) using KumoRFM-2 API calls. The latter relies on official benchmarking scripts for setting the various KumoRFM-2 hyperparameters that vary from task to task (see Footnote 3 and Appendix B.3). For datasets with no available scripts (avito and trial), the reproduction value is set to the original.

[†] OpenRFM is excluded when calculating average AUROC and Rank because results are not available for all datasets. Note that public code is not available to produce results for the missing tasks.

*Table 6.* Results on Relbench-v1 regression tasks (MAE↓). Bold = best per column; underline = second best.

| | avito | trial | | amazon | | stack | hm | Avg MAE | Avg Rank |
|---|---|---|---|---|---|---|---|---|---|
| **Method** | ctr | adverse | success | user | item | votes | sales | ↓ | ↓ |
| **Base** | | | | | | | | | |
| Global Median | 0.043 | 57.533 | 0.462 | 16.783 | 64.234 | 0.068 | 0.076 | 19.886 | 11.71 |
| Entity Median | 0.046 | 57.930 | 0.441 | 17.423 | 66.436 | 0.069 | 0.078 | 20.346 | 12.71 |
| **Supervised** | | | | | | | | | |
| LightGBM | 0.041 | 44.011 | 0.425 | 16.783 | 60.569 | 0.068 | 0.076 | 17.425 | 9.93 |
| GraphSAGE | 0.041 | 44.473 | 0.400 | 14.313 | 50.053 | 0.065 | 0.056 | 15.629 | 7.21 |
| RelGT | 0.035 | 43.992 | 0.326 | 14.267 | 48.922 | 0.065 | 0.054 | 15.380 | 5.14 |
| RelGNN | 0.037 | 44.461 | **0.301** | 14.230 | 48.767 | 0.065 | 0.054 | 15.416 | 5.21 |
| RelAgent | 0.033 | **37.194** | 0.386 | 13.949 | **41.765** | **0.064** | 0.035 | **13.347** | **2.21** |
| **Foundation Model** | | | | | | | | | |
| *Closed-Source* | | | | | | | | | |
| KumoRFM-1 | 0.035 | 58.231 | 0.417 | 16.161 | 55.254 | 0.065 | 0.040 | 18.600 | 7.64 |
| KumoRFM-2 (orig.)[*] | 0.034 | 43.293 | 0.433 | **13.921** | 46.992 | **0.064** | **0.034** | 14.967 | 3.86 |
| KumoRFM-2 (repro.)[*] | 0.033 | 41.974 | 0.433 | 14.627 | 45.352 | 0.065 | 0.043 | 14.647 | 4.93 |
| TabPFN-REL | **0.031** | 40.202 | 0.385 | 14.359 | 46.199 | 0.068 | 0.059 | 14.472 | 4.86 |
| *Open-Source* | | | | | | | | | |
| Griffin | 0.050 | 78.232 | 0.463 | 35.590 | 53.214 | 0.092 | 0.151 | 23.970 | 13.86 |
| RT | 0.058 | 73.999 | 0.455 | 18.802 | 57.996 | 0.110 | 0.089 | 21.644 | 13.71 |
| RandGNN | 0.038 | 53.370 | 0.434 | 17.580 | 66.130 | 0.071 | 0.065 | 19.670 | 11.43 |
| RDBLearn (v1.1) | 0.033 | 43.200 | 0.378 | 14.504 | 47.038 | 0.067 | 0.063 | 15.040 | 5.57 |

[*] See comment from Table 5.

# D. Technical Proofs

## D.1. Proof of Proposition 3.1

**High-Level Intuition.** To illustrate the gist of this result, assume that the ground-truth label $y_{\text{test}}$ for any $\boldsymbol{x}_{\text{test}}$ is given by a parity check on available labels in $\mathcal{G}_H^*(\boldsymbol{x}_{\text{test}})$. This labeling rule is impossible to determine from $y_H(\boldsymbol{x}_{\text{test}})$ alone, since for any node $\boldsymbol{x}'$ associated with a label $y' \in y_H(\boldsymbol{x}_{\text{test}})$, the corresponding set $y_H(\boldsymbol{x}')$ need not be fully observed, i.e., the overlap between $y_H(\boldsymbol{x}')$ (with a known label for $\boldsymbol{x}'$) and $y_H(\boldsymbol{x}_{\text{test}})$ (with an unknown label for $\boldsymbol{x}_{\text{test}}$) is only partial. Hence the parity rule is not identifiable, as even a single missing label can flip the result. This disrupts inductive inference from the information available within $\mathcal{G}_H^*(\boldsymbol{x}_{\text{test}})$, which is all a fixed FM encoder has access too. In contrast, if per-dataset training is allowed over RDB instances formed via a given parity-based labeling rule, it is straightforward to learn the associated mapping on a dataset-by-dataset or task-by-task basis. We remark that, although the proof below relies on parity checks for illustrative convenience, the underlying phenomena is emblematic of quite broad scenarios where diverse heterophilic (as opposed to homophilic) relationships are dominant.

**Basic Setup.** The basic proof follows from the construction of a suitable joint distribution over context $\widetilde{\mathcal{G}}_H^*(\boldsymbol{x}_{\text{test}})$ and test label $y_{\text{test}}$. Meanwhile, by assumption we have that $\mathcal{G}_H^*(\boldsymbol{x}_{\text{test}})$ is fixed and given. To begin, we introduce a latent indicator variable $z \in \{0, 1\}$ that selects among two deterministic generative processes for $\{\widetilde{\mathcal{G}}_H^*(\boldsymbol{x}_{\text{test}}), y_{\text{test}}\}$. In this way, all randomness in the joint distribution $p(\widetilde{\mathcal{G}}_H^*(\boldsymbol{x}_{\text{test}}), y_{\text{test}} \mid \mathcal{G}_H^*(\boldsymbol{x}_{\text{test}}))$ will ultimately reduce to randomness from $p(z)$. We emphasize that this is not meant to exclude broader sources of randomness that naturally describe RDB content; instead, we are merely distilling down to a minimal design case that nonetheless still allows us to establish the proposition. Other latent sources of variation associated with real-world RDBs then become largely orthogonal to our result and need not be explicitly specified.

**Label Assumptions.** We assume that all labels are binary, namely, $y \in \{0, 1\}$. While generalization to broader label domains $\mathcal{Y}$ is feasible, it introduces superfluous complexity given that the binary case is already sufficient for establishing the proof. For any $y_H(\boldsymbol{x}) \neq \emptyset$, where $y_H(\boldsymbol{x})$ indicates the set of all past labels present within $\mathcal{G}_H^*(\boldsymbol{x})$ for some arbitrary $\boldsymbol{x}$, we define conditional label generation via a parity function and its negative given by

$$
\begin{aligned}
\pi\big[y_H(\boldsymbol{x}) \, ; \, z = 1\big] &:= \left[\sum_{y \in y_H(\boldsymbol{x})} y\right] \bmod 2 \\
\pi\big[y_H(\boldsymbol{x}) \, ; \, z = 0\big] &:= \neg\pi\big[y_H(\boldsymbol{x}); z = 1\big],
\end{aligned}
\tag{2}
$$

both of which output a label $y \in \{0, 1\}$.

**Context Assumptions.** We specify a function $h\big(\mathcal{G}_H^*(\boldsymbol{x}_{\text{test}}); z\big)$ for context generation. To ensure no conflicting labels, for $z = 1$ we define this $h$ such that $\widetilde{\mathcal{G}}_H^*(\boldsymbol{x}_{\text{test}})$ has label assignments consistent with both $y_H(\boldsymbol{x}_{\text{test}})$ and the label assignment rule $\pi\big[\,\cdot\,; z = 1\big]$ from (2). This will always be achievable by introducing new appropriately-labeled neighbors to specific nodes of $\mathcal{G}_H^*(\boldsymbol{x}_{\text{test}})$ associated with elements of $y_H(\boldsymbol{x}_{\text{test}})$.

To establish this possibility, assume an arbitrary trial arrangement of node features, edges, and labels within $\widetilde{\mathcal{G}}_H^*(\boldsymbol{x}_{\text{test}})$, but following the typing of $\mathcal{G}_H^*(\boldsymbol{x}_{\text{test}})$ for consistency. Then consider any $y' \in y_H(\boldsymbol{x}_{\text{test}})$ with corresponding $\boldsymbol{x}'$ extracted from the associated row of $\boldsymbol{X} \equiv \boldsymbol{T}^K$ (per original definitions in the main text). For consistency with the labeling rule (2) with $z = 1$, we require that $y' = \pi\big[y_H(\boldsymbol{x}') \, ; \, z = 1\big]$, which may not initially be the case since labeling assignments were thus far assumed arbitrary. However, for any node where this is not the case, we can add a single additional labeled neighbor to $\widetilde{\mathcal{G}}_H^*(\boldsymbol{x}_{\text{test}})$, with an earlier time-stamp and a label selected to flip the parity check, such that we obtain the necessary equality.

Such a neighbor can be added in multiple ways given that $H \geq 2$ (note that if $H = 1$ then $|y_H(\boldsymbol{x}_{\text{test}})| = 0$, i.e., no labels within one hop, which is not allowed by assumption). First, assume $H = 2$, if some table $\boldsymbol{T}^k$ exists with at least two FK columns pointing to the PK column associated with $\boldsymbol{T}^K \equiv \boldsymbol{X}$, we can trivially add such a neighbor by adding a new labeled row to $\boldsymbol{T}^K$ and a new row to $\boldsymbol{T}^k$ that links the former to $\boldsymbol{x}' \in \boldsymbol{T}^K$. Meanwhile, if such a $\boldsymbol{T}^k$ does not exist, we can simply add such a table to $\widetilde{\mathcal{G}}_H^*(\boldsymbol{x}_{\text{test}})$. Moreover, when supplied with an earlier time-stamp, the added labeled node/row need not introduce any additional constraints vis-à-vis (2). With regard to the latter, if $\boldsymbol{x}''$ indicates the added node, we can directly enforce $y_H(\boldsymbol{x}'') = \emptyset$ as no nodes with earlier time-steps exist, and so $y''$ can be selected free of any constraint from (2), which would only apply if $y_H(\boldsymbol{x}'') \neq \emptyset$. Lastly, because this added label will by design be two hops from $\boldsymbol{x}'$ and three

or more hops from any other labeled rows, it will not introduce new conflicts per (2) if $H = 2$. For $H > 2$ an analogous procedure can be applied pushing the new labeled node to $H$ hops away from $\boldsymbol{x}'$ so as to achieve the same effect.

In an analogous manner we can also define $h$ when $z = 0$, handling label consistency via the introduction of neighboring nodes to adjust parity accordingly. Again, we are granted this flexibility by assumption, since only $\mathcal{G}_H^*(\boldsymbol{x}_{\text{test}})$ is fixed and given.

**Finalizing Part 1.** Per the assumptions stated above and the appropriate counting measure, it directly follows that $\int p\left(y_{\text{test}} | \mathcal{G}_{\text{test}}^*\right) \log p\left(y_{\text{test}} | \mathcal{G}_{\text{test}}^*\right) dy_{\text{test}} = 0$ trivially establishing that $y_{\text{test}}$ is a deterministic function of $\mathcal{G}_{\text{test}}^*$. Furthermore, given $\mathcal{G}_{\text{test}}^* = \left\{ \widetilde{\mathcal{G}}_H^*(\boldsymbol{x}_{\text{test}}), \mathcal{G}_H^*(\boldsymbol{x}_{\text{test}}) \right\}$ by definition, we also have that

$$
\mathbb{E}_{p\left(\widetilde{\mathcal{G}}_H^*(\boldsymbol{x}_{\text{test}}) | \mathcal{G}_H^*(\boldsymbol{x}_{\text{test}})\right)} \left[ \mathbb{KL}\left[ p(y_{\text{test}} | \mathcal{G}_{\text{test}}^*) \,||\, q_\theta\left(y_{\text{test}} \mid g_\phi\left[\mathcal{G}_H^*(\boldsymbol{x}_{\text{test}})\right]\right)\right]\right]
$$

$$
= \int p\left(\widetilde{\mathcal{G}}_H^*(\boldsymbol{x}_{\text{test}}) | \mathcal{G}_H^*(\boldsymbol{x}_{\text{test}})\right) \int p\left(y_{\text{test}} | \mathcal{G}_{\text{test}}^*\right) \log \frac{p\left(y_{\text{test}} | \mathcal{G}_{\text{test}}^*\right)}{q_\theta\left(y_{\text{test}} \mid g_\phi\left[\mathcal{G}_H^*(\boldsymbol{x}_{\text{test}})\right]\right)} dy_{\text{test}} d\widetilde{\mathcal{G}}_H^*(\boldsymbol{x}_{\text{test}})
$$

$$
= -\int p\left(\widetilde{\mathcal{G}}_H^*(\boldsymbol{x}_{\text{test}}) | \mathcal{G}_H^*(\boldsymbol{x}_{\text{test}})\right) \int p\left(y_{\text{test}} | \widetilde{\mathcal{G}}_H^*(\boldsymbol{x}_{\text{test}}), \mathcal{G}_H^*(\boldsymbol{x}_{\text{test}})\right) \log q_\theta\left(y_{\text{test}} \mid g_\phi\left[\mathcal{G}_H^*(\boldsymbol{x}_{\text{test}})\right]\right) dy_{\text{test}} d\widetilde{\mathcal{G}}_H^*(\boldsymbol{x}_{\text{test}})
$$

$$
= -\int \int p\left(\widetilde{\mathcal{G}}_H^*(\boldsymbol{x}_{\text{test}}), y_{\text{test}} | \mathcal{G}_H^*(\boldsymbol{x}_{\text{test}})\right) \log q_\theta\left(y_{\text{test}} \mid g_\phi\left[\mathcal{G}_H^*(\boldsymbol{x}_{\text{test}})\right]\right) dy_{\text{test}} d\widetilde{\mathcal{G}}_H^*(\boldsymbol{x}_{\text{test}}) \tag{3}
$$

for any fixed $g_\phi$ and $q_\theta$. By construction, $p\left(\widetilde{\mathcal{G}}_H^*(\boldsymbol{x}_{\text{test}}), y_{\text{test}} | \mathcal{G}_H^*(\boldsymbol{x}_{\text{test}})\right) \neq 0$ at only two points associated with $z = 0$ and $z = 1$. Let $y_{\text{test}}^{(0)}$ and $y_{\text{test}}^{(1)}$ denote the value of $y_{\text{test}}$ at these two points. Moreover, if we assume that $z$ is drawn uniformly over $\{0, 1\}$, we have $\mathcal{G}_H^*(\boldsymbol{x}_{\text{test}}) \perp\!\!\!\perp z$ and so we may conclude from (3) that

$$
\mathbb{E}_{p\left(\widetilde{\mathcal{G}}_H^*(\boldsymbol{x}_{\text{test}}) | \mathcal{G}_H^*(\boldsymbol{x}_{\text{test}})\right)} \left[ \mathbb{KL}\left[ p(y_{\text{test}} | \mathcal{G}_{\text{test}}) \,||\, q_\theta\left(y_{\text{test}} \mid g_\phi\left[\mathcal{G}_H^*(\boldsymbol{x}_{\text{test}})\right]\right)\right]\right]
$$

$$
= -p(z = 0) \log q_\theta\left(y_{\text{test}}^{(0)} \mid g_\phi\left[\mathcal{G}_H^*(\boldsymbol{x}_{\text{test}})\right]\right) - p(z = 1) \log q_\theta\left(y_{\text{test}}^{(1)} \mid g_\phi\left[\mathcal{G}_H^*(\boldsymbol{x}_{\text{test}})\right]\right)
$$

$$
\geq \mathbb{H}[z] = \log 2. \tag{4}
$$

This lower bound is achievable iff $q_\theta\left(y_{\text{test}} \mid g_\phi\left[\mathcal{G}_H^*(\boldsymbol{x}_{\text{test}})\right]\right)$ is a uniform distribution over $\{0, 1\}$.

**Finalizing Part 2.** Assume that some parameter-free encoder $g$ computes the element-wise sum of $y_H(\boldsymbol{x})$; other arbitrary features may be computed as well, but they are are ultimately treated as spurious features in what follows. Given this setup, the prediction head $q_\theta$ need only learn to differentiate even and odd sums associated with $y_H(\boldsymbol{x})$, while ignoring all other feature columns. For example, if $z = 1$ governs the generative process, then odd values of any sum over $y_H(\boldsymbol{x})$ will be mapped to 1, even values to 0. Such functions can be directly represented by ReLU networks, and while not a requirement for this proof, are even efficiently learnable with additional assumptions (Daniely & Malach, 2020).

### D.2. Proof of Proposition 4.1

**High-Level Intuition.** The proof follows through modifications applied to the original parity setup from Appendix D.1. For illustrative simplicity, assume a single informative binary RDB feature column, whereby ground-truth labels are a deterministic function, akin to a parity check, on the values of all features from this column contained within $\mathcal{G}_H^*(\boldsymbol{x}_{\text{test}})$. All other RDB columns are also binary but uninformative. Inclusion of $y_H(\boldsymbol{x}_{\text{test}})$ within $\mathcal{G}_H^*(\boldsymbol{x}_{\text{test}})$ does not allow for differentiating which column is informative, for analogous reasons to those presented in establishing Proposition 3.1. In brief, for any node $\boldsymbol{x}'$ associated with a label $y' \in y_H(\boldsymbol{x}_{\text{test}})$, the corresponding set $y_H(\boldsymbol{x}')$ *and its associated features* need not be fully observed, and so a parity function on *any* column can produce an arbitrary distribution of $y_H(\boldsymbol{x}_{\text{test}})$. Consequently, the parity rule is not identifiable regardless of whether $y_H(\boldsymbol{x}_{\text{test}})$ is present or not as an encoder input. Note that this line of reasoning is not restricted to binary feature columns, as we can quantize continuous or categorical columns into binary ones in forming deterministic labeling functions for analysis purposes. We now turn to the actual proof.

**Basic Setup.** As in the proof of Proposition 3.1, all randomness is encapsulated by a binary latent variable $z$ that selects between two deterministic generative processes, now over $\{\widetilde{\mathcal{G}}_H^*(\boldsymbol{x}_{\text{test}}), y_H(\boldsymbol{x}_{\text{test}}), y_{\text{test}}\}$, i.e., only $\mathcal{G}_H(\boldsymbol{x}_{\text{test}})$ as opposed to

$\mathcal{G}_H^*(\boldsymbol{x}_{\text{test}})$ is assumed fixed/given. When granted $\mathcal{G}_H(\boldsymbol{x})$ for some arbitrary $\boldsymbol{x}$, we specify the conditional label generation functions

$$
\begin{aligned}
\pi\big[\mathcal{G}_H(\boldsymbol{x})\,;\; z = 1\big] &:= \left\lfloor \sum_{\{i,j,k\} \in S_H^{(1)}(\boldsymbol{x})} \beta\left(t_{ij}^k\right) \right\rfloor \bmod 2 \\
\pi\big[\mathcal{G}_H(\boldsymbol{x})\,;\; z = 0\big] &:= \left\lfloor \sum_{\{i,j,k\} \in S_H^{(0)}(\boldsymbol{x})} \beta\left(t_{ij}^k\right) \right\rfloor \bmod 2,
\end{aligned}
\tag{5}
$$

where $S_H^{(1)}(\boldsymbol{x})$ and $S_H^{(0)}(\boldsymbol{x})$, associated with $z = 1$ and $z = 0$ respectively, represent distinct support sets over RDB column elements reachable within $H$ hops from $\boldsymbol{x}$ following PK-FK pairs. In this way any $\{i, j, k\} \in S_H^{(1)}(\boldsymbol{x})$ (or analogously, within $S_H^{(0)}(\boldsymbol{x})$) indicates the $i$-th element within a specific RDB column $\boldsymbol{t}_{:j}^k$, namely column $j$ in table $k$. Meanwhile $\beta$ denotes a mapping from each table element to a binary value (this discretization is by no means necessary, but it facilitates a simplified exposition that is sufficient for establishing the core result). These values are then subsequently combined via the stated parity functions in (5) to produce binary labels $y \in \{0, 1\}$.

**Establishing Part 1.** By adopting (5) as the label generating function, we trivially achieve the conditions of part 1 in Proposition 4.1. Specifically, with all randomness deferred to $z$, which explicitly fixes the support set, all labels default to a deterministic function of the chosen support set, either $S_H^{(0)}(\boldsymbol{x})$ or $S_H^{(1)}(\boldsymbol{x})$ for any $\boldsymbol{x}$.

**Establishing Part 2.** If we stipulate that $z$ is drawn uniformly over $\{0, 1\}$ independently of $\mathcal{G}_H(\boldsymbol{x}_{\text{test}})$, then we have

$$
p\left(S = S_H^{(0)}(\boldsymbol{x}_{\text{test}}) \mid \mathcal{G}_H(\boldsymbol{x}_{\text{test}})\right) = p\left(S = S_H^{(1)}(\boldsymbol{x}_{\text{test}}) \mid \mathcal{G}_H(\boldsymbol{x}_{\text{test}})\right) = \tfrac{1}{2}.
\tag{6}
$$

However, by design we also have the flexibility to select the distribution over $\widetilde{\mathcal{G}}_H^*(\boldsymbol{x}_{\text{test}})$ such that $y_H(\boldsymbol{x}_{\text{test}})$ is invariant to $z$, even while $y_{\text{test}}$ still depends on $z$ through (5) applied at $\boldsymbol{x}_{\text{test}}$. To see this, note that each $y' \in y_H(\boldsymbol{x}_{\text{test}})$ is computed as $\pi[\mathcal{G}_H(\boldsymbol{x}'); z]$, where $\boldsymbol{x}'$ is the target table feature row associated with $y'$. Hence we only need enforce

$$
\pi\big[\mathcal{G}_H(\boldsymbol{x}');\; z = 1\big] = \pi\big[\mathcal{G}_H(\boldsymbol{x}');\; z = 0\big] \quad \text{for each } y' \in y_H(\boldsymbol{x}_{\text{test}}).
\tag{7}
$$

But this will always be possible per the following: Suppose the equality from (7) does not initially hold. We may add an additional element to either $S_H^{(1)}(\boldsymbol{x}')$ or $S_H^{(0)}(\boldsymbol{x}')$ (but not both) such that the corresponding tuple produces $\beta(t_{ij}^k) = 1$ to flip the parity and induce equality. This is achievable without disrupting equality elsewhere by placing the added element within $\widetilde{\mathcal{G}}_H^*(\boldsymbol{x}_{\text{test}})$, at a location $H$ hops from $\boldsymbol{x}'$ and greater than $H$ hops from other elements of $y_H(\boldsymbol{x}_{\text{test}})$. This placement mirrors an analogous placement strategy applied in Appendix D.1, only here labeled nodes are not required.

Lastly, if (7) holds, then

$$
p\big(S|\mathcal{G}_H^*(\boldsymbol{x}_{\text{test}})\big) = p\big(S|\mathcal{G}_H(\boldsymbol{x}_{\text{test}}), y_H(\boldsymbol{x}_{\text{test}})\big) = p\big(S|\mathcal{G}_H(\boldsymbol{x}_{\text{test}})\big),
\tag{8}
$$

completing the demonstration of part 2.

