# OpenReview forum: "Parameter-Free Encoders Remain Viable for RDB Foundation Models"
_ICML.cc/2026/Workshop/FMSD — FMSD @ ICML 2026 Poster_

### Official Review · Reviewer_iY76 · 2026-05-20
**Sharp position paper with credible theory and competitive empirics, but narrow benchmark scope and reliance on closed-source comparisons limit the strength of the claim.**

**Rating:** 7
**Confidence:** 3

**Review:**

Strengths:

1. Clean, well-scoped argument: two informal propositions establish identifiability limits for fixed (foundation-model) encoders that consume neighborhood labels, with proof sketches (parity-based counterexamples) that are intuitive and directly tied to the empirical question of whether parameterized RDB encoders are worth the pre-training cost.

2. Empirical results are honest and self-aware: RDBLearn (parameter-free) tops the ranking on 5 RelBench-v2 tasks and 5 4DBInfer tasks against KumoRFM-1/2 and Griffin, and the appendix openly discusses the closed-source nature of KumoRFM-2 and a plausible train/dev-set-merging confound that narrows the gap further.

Weaknesses:
1. The propositions are stated only in "informal" form with proof sketches and the formal versions are deferred to an unnamed "longer technical report" - for a theory-flavored position paper this is a real gap, since the strength of the claim hinges on what "representative ego-network" and "general position" actually mean formally.
2. Empirical evidence is thinner than the framing suggests: only 5 RelBench-v2 tasks (all from a single dataset, rel-RateBeer, after excluding rel-mimic and rel-arXiv) plus 5 4DBInfer classification tasks; no regression on 4DBInfer, no significance testing, and the main competitors (KumoRFM-1/2) are closed-source, so the comparison cannot be fully audited.

---

### Official Review · Reviewer_4RSR · 2026-05-22
**Insightful work and good performance; but connection to real-world settings is missing.**

**Rating:** 7
**Confidence:** 4

**Review:**

The paper shows that simple parameter-free relational encoders combined with frozen table foundation models can match state-of-the-art performance for prediction tasks in relational databases. It also proves that trainable encoder parameters offer limited benefit when labels are already included in the inputs.

The paper does not suggest anything surprisingly novel, but the research is impactful. Specifically, their model outperforming KumoFM-2 is particularly impressive, as it clearly shows that simpler techniques can improve over SOTA models. Overall, it is a good and insightful paper.

One major aspect the paper lacks is application to real-world settings, as the evaluation mainly relies on benchmark datasets and the theory is based on simplified artificial examples.

---

### Official Review · Reviewer_NyuJ · 2026-05-22
**Review for ICML 2026 Workshop FMSD Submission200.**

**Rating:** 7
**Confidence:** 5

**Review:**

- Summary: The paper examines the possibility of parameter-free RDB encoding combined with single-table foundation models for predictive machine leaning.

- Strengths: The paper is easy to follow, and the topic is important in the field of tabular learning and the corresponding foundation models.

- Detailed Comments: Despite the spacing concerns, it might be useful to have a small figure explaining the 'Labels as Extra Discriminative Features' for better understanding. Also, for the empirical results, it would be nice to have some statistical testings (although it might be difficult due to performance reports for respective models).

- Justification of score: Overall, the paper holds important insights that should be explored more in depth, introducing new possibilities.